# Transcriptome Analysis of the Regulatory Mechanism of Exogenous Spermidine in High Temperature Stress Resistance of Tomato Seedlings

Chen Peng [1,2], Sheng Shu [1], Yu Wang [1], Jing Du [1], Lu Shi [1,2], Mohammad Shah Jahan [1] and Shirong Guo [1,*]

[1] College of Horticulture, Nanjing Agricultural University, Key Laboratory of Southern Vegetable Crop Genetic Improvement, Ministry of Agriculture, Nanjing 210095, China
[2] Jiangsu Academy of Agricultural Sciences, Nanjing 210014, China
[*] Correspondence: srguo@njau.edu.cn; Tel.: +86-025-8439-5267

**Abstract:** Previous studies have shown that spermidine (Spd) can improve tolerance to high temperature stress in tomato seedlings. To further understand how Spd regulates the molecular components of high temperature stress signaling pathways, we performed a genome-wide transcriptome analysis in tomato seedlings treated with high temperature and/or exogenous Spd. The results demonstrate that, under high temperature conditions, Spd significantly alleviated the inhibition of plant growth, as well as improving the net photosynthetic rate and pigment contents. The transcriptome analysis revealed thousands of differentially expressed genes (DEGs) in response to high temperature with or without Spd treatment. Half of the genes were induced by high temperature, part of the genes were induced by high temperature with exogenous Spd, and some were induced by the coordinated effect of high temperature and Spd. A GO analysis indicated that genes involved in cellular processes, metabolic processes, and nucleotide binding in the sample were subjected to high temperature. Some DEGs were also involved in plant physiological processes. These results suggest potential genes and molecular pathways were involved in the exogenous Spd-mediated tolerance to high temperature stress in tomato plants. A JA signaling test was designed, which indicated that MYC2 and JAS1 in heat-resistant materials were both increased, through quantitative RT-PCR.

**Keywords:** high temperature stress; exogenous spermidine; transcriptome

## 1. Introduction

Global warming is predicted to have a generally negative effect on plant growth, due to the damaging effect of high temperatures on plant development. The increasing threat of high temperatures might lead to a loss of crop productivity [1]. Heat stress has an independent mode of action on the physiological metabolism of plant cells. Correspondingly, plants exhibit a characteristic set of cellular and metabolic responses required for survival under high temperature conditions [2]. The corresponding effects include changes in the organization of cellular structures [3], protein synthesis and accumulation [4,5], and the production of signaling molecules such as phytohormones, reactive oxygen species (ROS), and other protective molecules [6–10].

Polyamines (PAs) are unique low molecular weight polycationic aliphatic amines, including spermidine (Spd), putrescine (Put), and spermine (Spm), which can promote plant growth under both normal or abiotic stress conditions [11,12]. Recent studies have shown that PAs act as signaling molecules in the responses of plants to various abiotic stresses [13], through the activation of novel ion conductance, such as $Ca^{2+}$ influx across the plasma membrane (PM) [14]. PAs can help to detoxify the ROS that accumulate under various abiotic stresses, such as high temperature, salinity, and drought [11]. It has been reported that Spd plays an important role in plant growth and development [15,16]; however, only a few studies have focused on studying the transcriptome changes related

to the effects of Spd on plant responses to high temperature stress, and its regulatory mechanism remains unclear [17].

Many strategies have been applied to determine the key molecular components that enhance the high temperature stress tolerance of plants [18,19], including transcriptome sequencing, proteomic analysis [20], and metabolism assay [6]. Genome-wide expression analysis is an attractive approach to understanding complex traits such as high temperature stress tolerance. For genome-wide transcription profiling, high-throughput RNA sequencing (RNA-seq) is the best approach, which has replaced microarray analysis in plants, leading to the availability of high-quality whole-genome sequence information [21–23]. During recent years, through efforts based on RNA-seq, a large number of differentially expressed genes (DEGs) under high temperature stress treatments have been identified for many plants [21,24–30].

Tomato (*Lycopersicon esculentum*) is one of the most nutritional, economic, and widely consumed vegetables in the world. In Asia, tomato plants are grown in greenhouses, and are often exposed to high temperature stress in the summer [31]. Previous studies have shown, through proteomic analysis, that Spd can promote high temperature stress tolerance in tomato [32]; however, the proteomic analysis only identified a limited number of differentially expressed proteins in tomato to elucidate the effects of Spd on such high temperature tolerance. For further elaboration, in this study, RNA-seq analysis was conducted to identify the transcription profiles of responsive genes and global patterns in gene expression with or without exogenous Spd under high temperature stress in tomato.

## 2. Results

### 2.1. Plant Phenotyping and Physiological Reaction

#### 2.1.1. Plant Growth

After seven days of high temperature treatment, tomato seedlings showed significantly lower total dry weight and aerial part dry weight than those grown under normal conditions. However, these growth defects caused by high temperature could be restored through the addition of Spd (Figure 1A,B). High temperature did not affect the fruiting branching numbers, compared with normal conditions, while the Spd treatment helped to increase fruiting branching numbers under high temperature (Figure 1C). High temperature caused a strongly reduced fruit set rate, from 81% to 71% (Figure 1D), while the Spd treatment promoted the fruit set rate strongly, both under high and normal temperature conditions: The fruit set rate increased to 86% under high temperature and was even higher than that under normal conditions (Figure 1D).

#### 2.1.2. Soluble Proteins

Soluble protein contents are major products of metabolism. In plants, they act not only as an essential source of energy or enzymes, but also as signaling molecules in response to the integration of information from environmental signals as well as developmental cues. When stress signaling is recognized by receptors and transmitted to downstream networks, proteins are transported into and accumulate in the cytoplasm, serving as crucial factors in fruit development and stress tolerance. The results indicated that the accumulation of soluble protein content increased from 53 to 76 µg/g under high temperature stress and was also slightly increased in samples with Spd treatment in advance (Figure 2), indicating that Spd plays a positive role in stress tolerance.

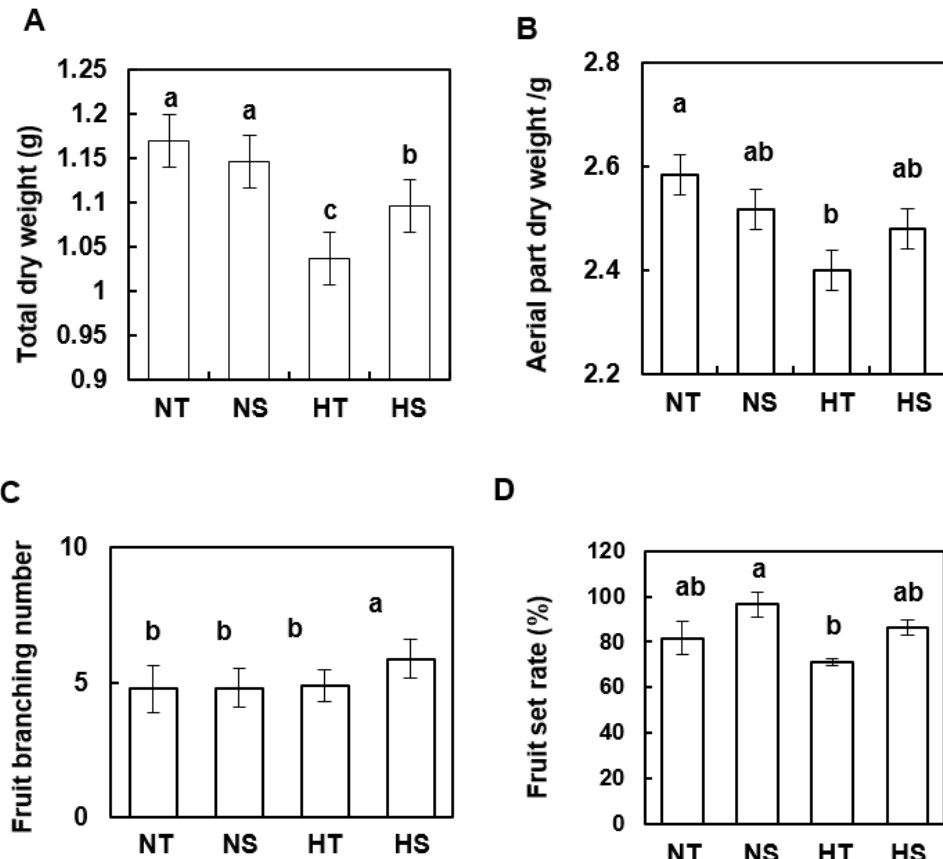

**Figure 1.** Effects of exogenous spermidine on biomass and fruit of tomato seedlings under high temperature stress: (**A**) Total dry weight; (**B**) Aerial part dry weight; (**C**) Fruit branching number; and (**D**) Fruit set rate. Note: NT, normal growth condition under 25/18 °C and water; NS, normal growth condition under 25/18 °C with 1 mM Spd treatment on plants; HT, high temperature growth condition on 38/28 °C and water treatment; HS, high temperature growth condition on 38/28 °C with 1 mM Spd treatment on leaves. Error bars represent SDs among three independent replicates. Different letters represent significant differences with $p < 0.05$ (Duncan's multiple range tests).

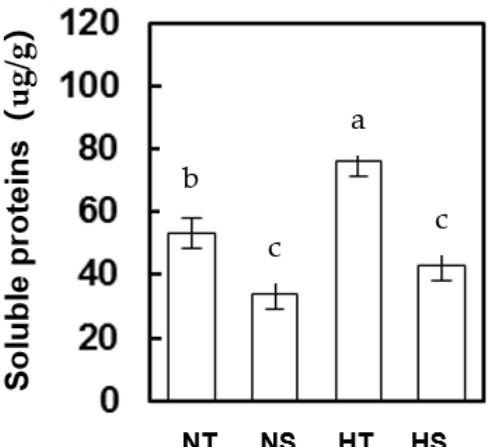

**Figure 2.** Accumulation of soluble proteins in tomato under high temperature stress with or without treatment of Spd. Note: Error bars represent SDs among three independent replicates. Values sharing the same letters are not significantly different ($p < 0.05$; Duncan's multiple range test). NT, normal growth condition under 25/18 °C and water; NS, normal growth condition under 25/18 °C with 1 mM Spd treatment on plants; HT, high temperature growth condition on 38/28 °C and water treatment; HS, high temperature growth condition on 38/28 °C with 1 mM Spd treatment on leaves.

### 2.1.3. Photosynthetic Parameters

Heat stress negatively affects respiration and photosynthesis in plant leaves, compromising the structure of chloroplast protein complexes, reducing enzyme activities, and causing injuries to the membrane system as well as the chlorophyll and photosynthetic apparatus [5,33–35]. High temperature stress significantly reduced the transpiration rate (*E*), net photosynthetic rate (*Pn*; to 50%), intercellular $CO_2$ concentration (*Ci*; to 65%), and stomatal conductance (*gws*; to 30%), compared with NT. Furthermore, the exogenous application of Spd did not alter any these stomatal parameters under normal conditions (Figure 3A–D) and, importantly, maintained the net photosynthetic rate and stomatal conductance in tomato leaves exposed to high temperature stress (Figure 3B,D).

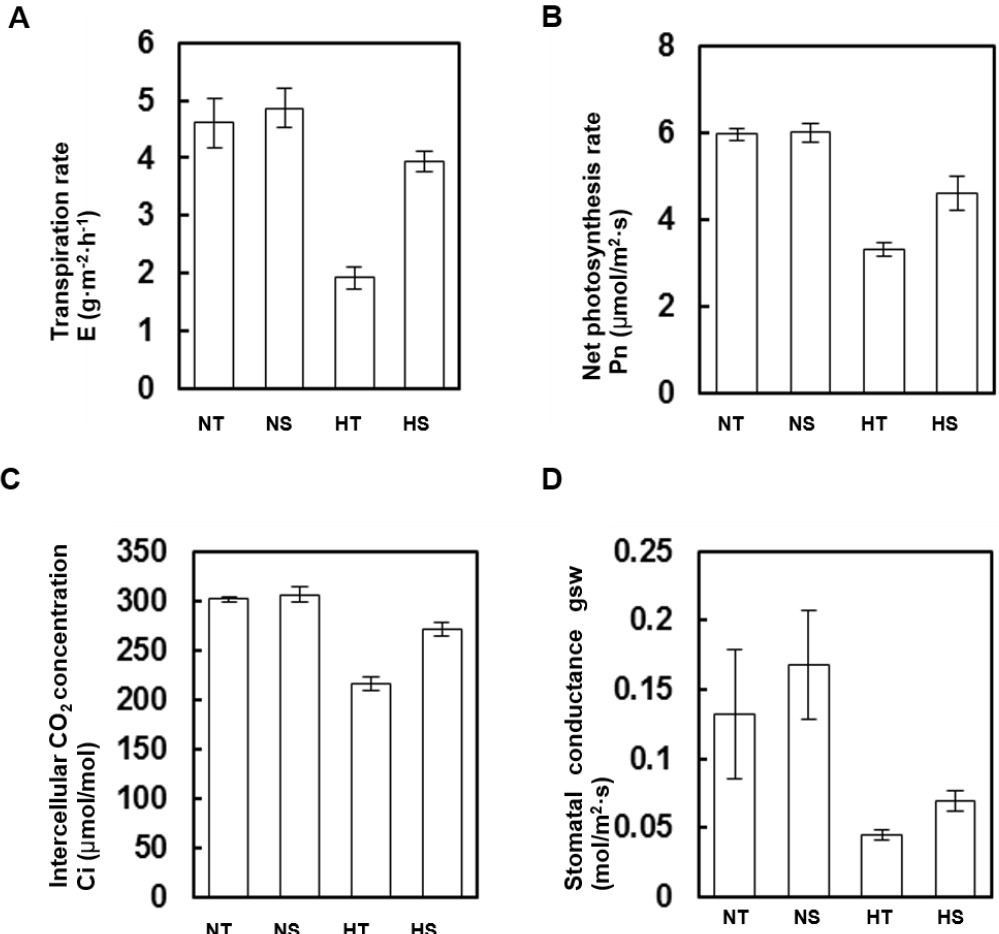

**Figure 3.** Effects of Spd on gas exchange parameters in tomato leaves under high temperature stress: Transpiration rate (**A**); net photosynthesis rate (**B**); intercellular $CO_2$ concentration (**C**); and stomatal conductance (**D**). NT, normal growth condition under 25/18 °C and water; NS, normal growth condition under 25/18 °C with 1 mM Spd treatment on plants; HT, high temperature growth condition on 38/28 °C and water treatment; HS, high temperature growth condition on 38/28 °C with 1 mM Spd treatment on leaves.

### 2.1.4. Chlorophyll Content

By increasing chlorophyllase (Chl) activity and decreasing the number of photosynthetic pigments, heat stress ultimately reduces the photosynthetic and respiratory activity in plants [33]. From our data, we found that high temperature stress significantly repressed the contents of Chl a, Chl b, total Chl a+b, and carotenoids; however, the application of exogenous Spd alleviated these adverse effects of high temperature stress on leaf photosynthetic pigment concentrations (Figure 4).

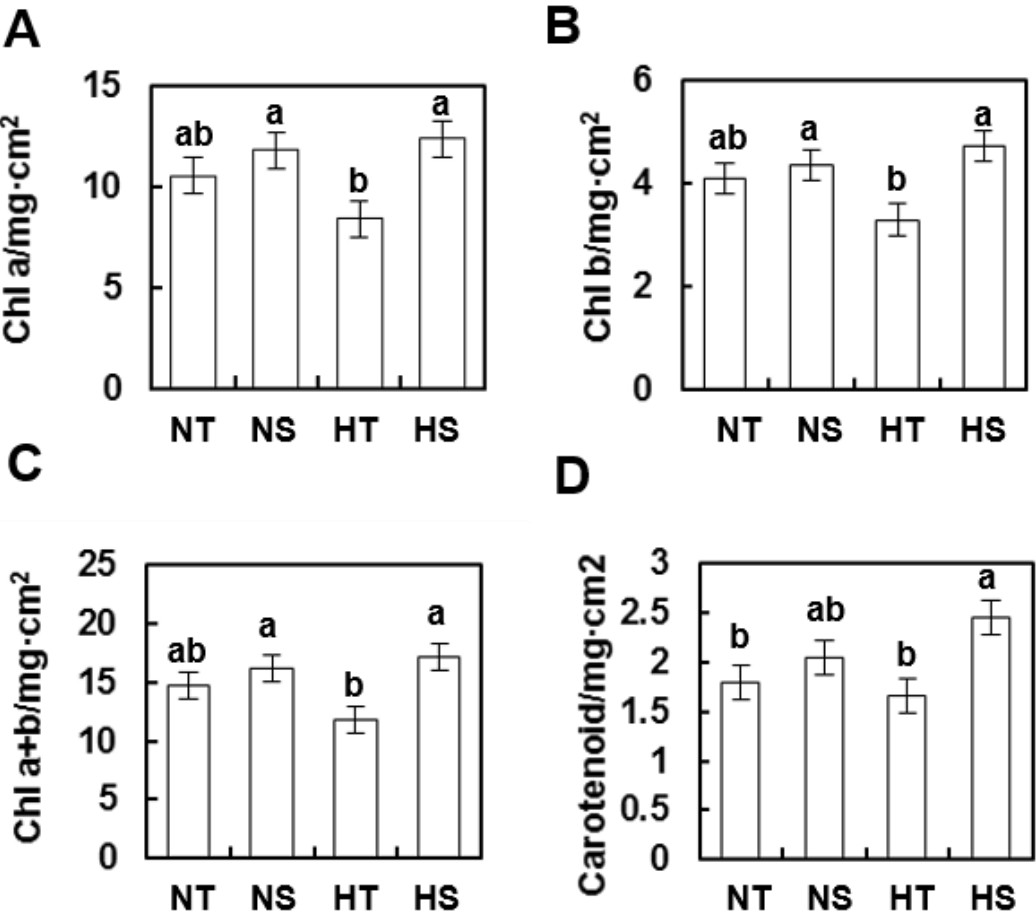

**Figure 4.** Photosynthetic pigments of tomato plants growing under normal conditions or high temperature, with or without exogenous application of Spd((**A**) is Chl a, (**B**) is Chl b, (**C**) is Chl a+b, (**D**) is carotenold). Error bars represent the SDs among three independent experiments. Note: Different letters represent significant differences with $p < 0.05$ (Duncan's multiple range tests). NT, normal growth condition under 25/18 °C and water; NS, normal growth condition under 25/18 °C with 1 mM Spd treatment on plants; HT, high temperature growth condition on 38/28 °C and water treatment; HS, high temperature growth condition on 38/28 °C with 1 mM Spd treatment on leaves.

*2.2. Transcriptome Analysis Results*

2.2.1. Summary of RNA Sequencing and Assembly

Four libraries were constructed to compare the gene expression changes occurring at the transcriptome level under high temperature and/or exogenous Spd. In total, approximately 636 million raw reads were obtained from the RNA-seq of the 12 samples, ranging from 45.08 to a maximum of 56.68 million reads from each library (Table 1). The sequencing reads containing low-quality, adaptor-polluted, and high content of unknown base (N) reads were removed before downstream analyses. A total of 60 GB, comprising 532.58 million clean reads were selected from the four libraries, with a minimum of 39.34 and a maximum of 45.26 million reads from each library. The Q30 (%) value of every library ranged from 88.85 to 90.84%, indicating the high quality of our sequencing (Table 1). After filtering, all of the clean reads were mapped to a reference genome using HISAT. On average, 83.04% of reads were mapped, and the uniformity of the mapping result for each library suggests that the libraries and repeats were comparable (Table 2).

**Table 1.** Statistics of sequencing reads after filtering.

| Library | Total Raw Reads (Mb) | Total Clean Reads (Mb) | Clean Reads Q30 (%) |
| --- | --- | --- | --- |
| NT-1 | 45.08 | 39.34 | 89.75 |
| NT-2 | 50.20 | 44.11 | 90.08 |
| NT-3 | 53.44 | 44.28 | 89.33 |
| NS-1 | 55.06 | 44.68 | 89.81 |
| NS-2 | 51.82 | 45.40 | 88.85 |
| NS-3 | 53.44 | 45.26 | 89.64 |
| HT-1 | 53.44 | 45.25 | 89.42 |
| HT-2 | 56.68 | 45.13 | 90.51 |
| HT-3 | 53.44 | 44.29 | 90.08 |
| HS-1 | 53.44 | 44.39 | 90.41 |
| HS-2 | 53.44 | 45.20 | 90.84 |
| HS-3 | 56.68 | 45.26 | 89.20 |

**Table 2.** Statistics of reads mapped onto tomato reference genomes.

| Sample | Total Clean Reads | Total Mapping Ratio | Uniquely Mapping Ratio |
| --- | --- | --- | --- |
| NT-1 | 39,338,682 | 81.52% | 79.17% |
| NT-2 | 44,112,738 | 83.55% | 81.25% |
| NT-3 | 44,277,524 | 83.55% | 81.11% |
| NS-1 | 44,684,888 | 82.85% | 80.38% |
| NS-2 | 45,400,480 | 82.35% | 79.92% |
| NS-3 | 45,260,080 | 82.81% | 80.24% |
| HT-1 | 45,250,174 | 81.55% | 79.69% |
| HT-2 | 45,133,642 | 83.37% | 81.55% |
| HT-3 | 44,287,642 | 83.57% | 81.79% |
| HS-1 | 44,388,334 | 84.44% | 82.39% |
| HS-2 | 45,196,132 | 83.06% | 81.06% |
| HS-3 | 45,262,252 | 83.91% | 81.95% |

2.2.2. Novel Transcript Prediction and SNP and INDEL Detection

After genome mapping, transcripts were reconstructed using StringTie. Then, with genome annotation information, novel transcripts in the samples were identified using cuffcompare (a tool in cufflinks). In total, 35,833 novel transcripts were identified, including 19,882 coding transcripts and 15,951 non-coding transcripts (Table 3).

**Table 3.** Summary of novel transcripts identified from all samples.

| Type of Novel Transcript | Number |
| --- | --- |
| Total Novel Transcript | 35,833 |
| Novel Coding Transcript | 19,882 |
| Novel Non-coding Transcript | 15,951 |
| Novel Isoform | 18,696 |
| Novel Gene | 1186 |

SNP and INDEL variants for each sample were identified using GATK. As shown in Table 4, approximately 20,000 SNP and INDEL variants were found in each sample. The numbers of 'A to G' and 'C to T' transition-type SNPs of each sample were accounted for, from a minimum of 11,753 to a maximum of 15,870. On average, there were about 8850 transversion-type SNPs, including A to C, A to T, C to G, and G to T, in each sample (Table 4). Further genome distribution analysis revealed that the SNP and INDEL sites were highly enriched in the exons, where they accounted for 55% and 47%, respectively, indicating large numbers of protein variants produced from these transcript variants (Figure 5A,B). DSG is regulated by alternative splicing (AS), which allows for the production of a variety of different isoforms from one gene only. Changes in the relative abundance of isoforms, regardless of the expression change, indicate a splicing-related mechanism.

Five types of AS events—including Skipped Exon, Alternative 5′ Splicing Site, Alternative 3′ Splicing Site, Mutually exclusive exons, and Retained Intron—were analyzed in our results (Figure 5C).

**Table 4.** Statistics of SNP variant type and number.

| Library | A–G | C–T | Transition | A–C | A–T | C–G | G–T | Transversion | Total |
|---------|-----|-----|------------|-----|-----|-----|-----|--------------|-------|
| HS-1 | 6434 | 6288 | 12,722 | 1881 | 2825 | 1297 | 1992 | 7995 | 20,717 |
| HS-2 | 7597 | 7449 | 15,046 | 2407 | 3493 | 1498 | 2503 | 9901 | 24,947 |
| HS-3 | 6696 | 6582 | 13,278 | 1966 | 2936 | 1343 | 2061 | 8306 | 21,584 |
| HT-1 | 7991 | 7879 | 15,870 | 2601 | 3831 | 1595 | 2638 | 10,665 | 26,535 |
| HT-2 | 7292 | 7119 | 14,411 | 2145 | 3102 | 1387 | 2252 | 8886 | 23,297 |
| HT-3 | 7220 | 7041 | 14,261 | 2142 | 2942 | 1449 | 2175 | 8708 | 22,969 |
| NS-1 | 6046 | 6077 | 12,123 | 1956 | 3137 | 1241 | 2101 | 8435 | 20,558 |
| NS-2 | 5945 | 5818 | 11,763 | 1851 | 2946 | 1212 | 1952 | 7961 | 19,724 |
| NS-3 | 6677 | 6480 | 13,157 | 2132 | 3479 | 1350 | 2292 | 9253 | 22,410 |
| NT-1 | 5932 | 5821 | 11,753 | 1862 | 3054 | 1229 | 1957 | 8102 | 19,855 |
| NT-2 | 6177 | 6167 | 12,344 | 1951 | 3021 | 1266 | 2010 | 8248 | 20,592 |
| NT-3 | 7272 | 7187 | 14,459 | 2348 | 3492 | 1465 | 2432 | 9737 | 24,196 |

Note: Transition, variant between purines or pyrimidines; Transversion, variant between purine and pyrimidine.

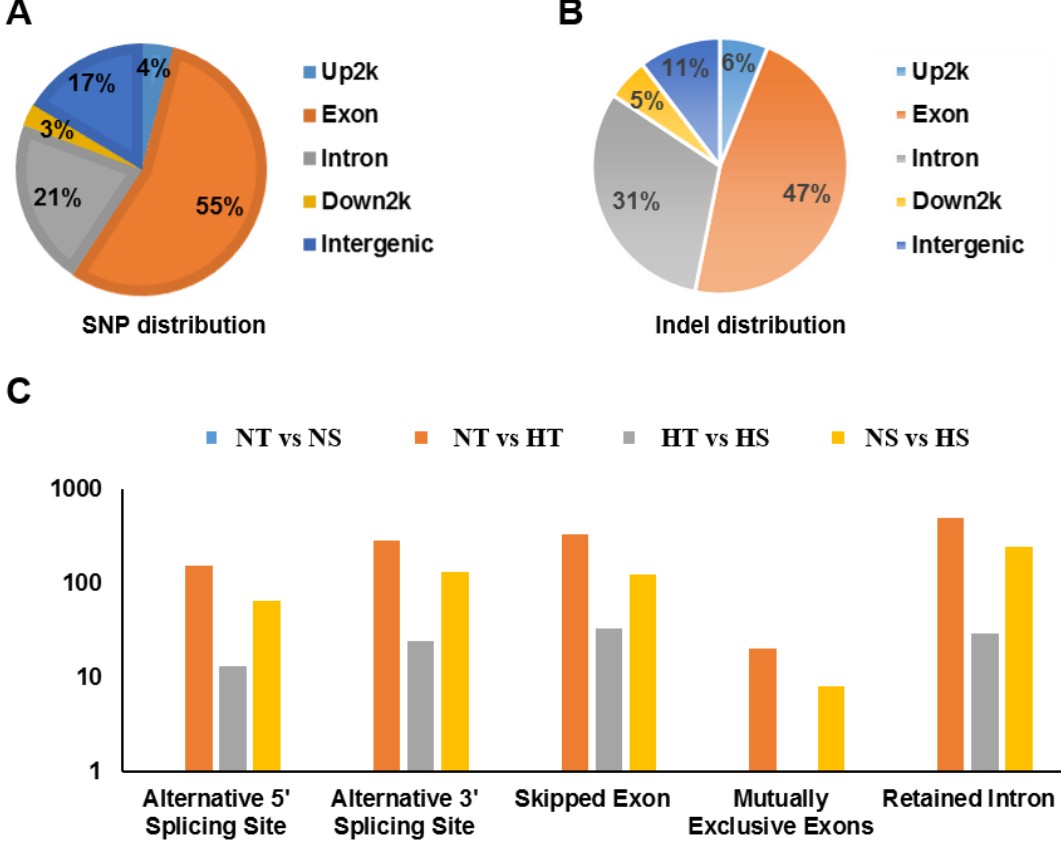

**Figure 5.** Quality of single nucleotide polymorphisms (SNPs), insertion/deletions (Indels), and alternative splicing sites in NT, NS, HT, and HS. NT, normal growth condition under 25/18 °C and water; NS, normal growth condition under 25/18 °C with 1 mM Spd treatment on plants; HT, high temperature growth condition on 38/28 °C and water treatment; HS, high temperature growth condition on 38/28 °C with 1 mM Spd treatment on leaves. Note: Relative distribution of SNPs (**A**) and Indels (**B**) in all data sets; (**C**) Numbers of alternative splicing sites (ASs) in five different types—alternative 5′splicing site, alternative 3′splicing site, skipped exon, mutually exclusive exons, and retained intron. Assays of ASs were performed between NT and NS, NT and HT, HT and HS, and NS and HS.

### 2.2.3. Gene Expression and Differentially Expressed Genes Analysis

After detection of novel transcripts, novel coding transcripts were merged with reference transcripts to obtain a complete reference. Then, the clean reads were mapped to the reference using Bowtie2, and the gene expression level for each sample was calculated with RSEM. On average, the total number of expressed genes ranged between 21,149 and 22,063 in all the samples, and the total number of transcripts ranged from 31,092 to 33,033. The read coverage and distribution on each detected transcript are provided in Supplementary Fiugre S1A,B, which indicated a high degree of mRNA integrity. Every two replicates shared a large number of expressed genes, covering more than 90% of one set (Supplementary Fiugre S1C). Moreover, the Pearson correlation indices between the replicated samples and the same treatment always exceeded 0.9, and were higher than the values between the different treatment samples (Figure 6A), in accordance with the hierarchical clustering between all samples (Figure 6B).

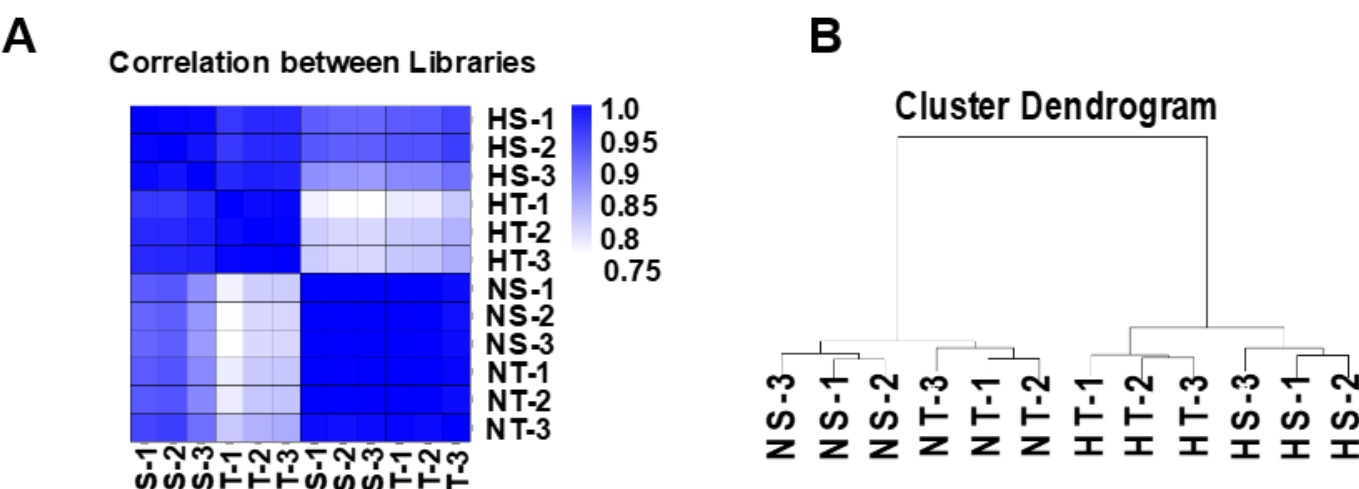

**Figure 6.** Summary of gene expression of each sample: (**A**) Pearson correlations between all samples; and (**B**) Hierarchical clustering between all samples. NT, normal growth condition under 25/18 °C and water; NS, normal growth condition under 25/18 °C with 1 mM Spd treatment on plants; HT, high temperature growth condition on 38/28 °C and water treatment; HS, high temperature growth condition on 38/28 °C with 1 mM Spd treatment on leaves.

For identification of differentially expressed genes from different pairs of biologically meaningful comparisons, a rigorous comparison at $p$-value $\leq 0.05$ and $\log_2$ fold change $\geq 2$ (for up-regulation) or $\leq -2$ (for down-regulation) was set. The genes synergistically or reversely regulated by Spd and high temperature might be important for Spd-mediated plant tolerance against the detrimental effects of heat. Under normal growth conditions, 24 DEGs were up-regulated and 28 DEGs were down-regulated with the Spd treatment. With just water treatment, 1741 DEGs were up-regulated and 1917 DEGs were down-regulated at a temperature of 38/28 °C. Under high temperature conditions, 288 DEGs were up-regulated and 235 DEGs were down-regulated after the Spd treatment. Meanwhile, with Spd spraying, 936 DEGs were up-regulated and 1038 DEGs were down-regulated under high temperature growth condition (Figure 7). A large number of DEGs existed in both samples with or without the exogenous Spd treatment under high temperature conditions, indicating the critical effects of high temperature stress.

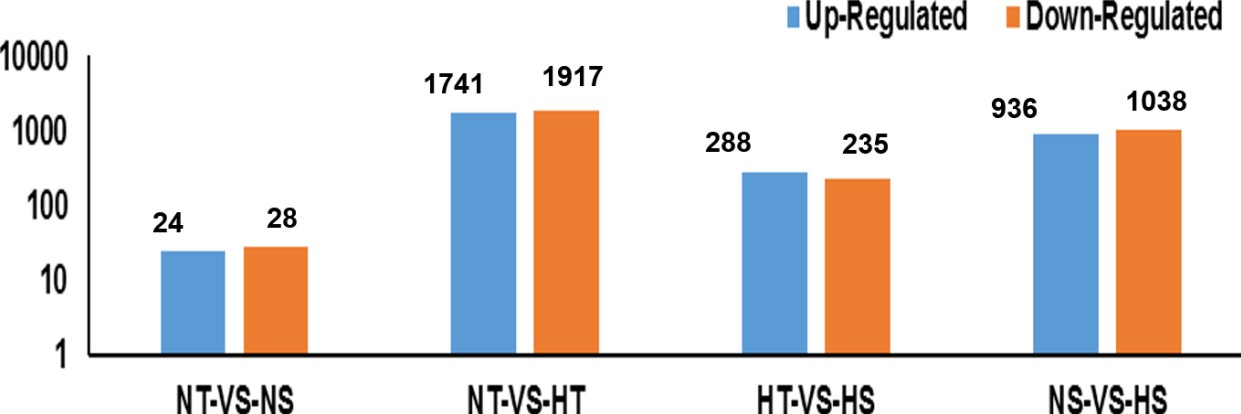

**Figure 7.** The number of differentially expressed genes (DEG) with or without Spd treatment under normal conditions or heat temperature stress. Summary of DEGs between NT and NS, NT and HT, HT and HS, and NS and HS. NT, normal growth condition under 25/18 °C and water; NS, normal growth condition under 25/18 °C with 1 mM Spd treatment on plants; HT, high temperature growth condition on 38/28 °C and water treatment; HS, high temperature growth condition on 38/28 °C with 1 mM Spd treatment on leaves.

Transcriptional factors (TFs), which regulate gene expression, rely on a DNA-binding domain (DBD) that recognizes a specific DNA sequence [36]. In our data, hundreds of TFs were significantly altered under high temperature stress. These TFs belong to different TF families, including *AP2/ERF, ARF, B3, BES1, C3H, TCP, bZIP, MADS, FAR1, C2H2, Dof, EIL, HD-ZIP, MYB, NAC, bHLH, GATA, HLH, mTERF*, and *WRKY*. There were 26 TFs induced by high temperature stress, 4 of which were down-regulated after treatment with Spd, including *Solyc07g040680, Solyc06g062460, Solyc08g075950*, and *Solyc02g069320*. Moreover, a group of 19 genes were down-regulated or unchanged under high temperature stress but were up-regulated when treated with both high temperature and the exogenous Spd. Among them, there were two MYB TFs (*Solyc12g098370* and *Solyc06g083900*), and four WRKY TFs (*Solyc03g116890, Solyc02g032950, Solyc07g055280,* and *Solyc08g082110*). Furthermore, interestingly, several key components of the JA signaling pathway, such as *Solyc01g096370* (*MYC2*), *Solyc12g009220* (*JAZ1*), *Solyc08g036640* (*JAZ5*), *Solyc12g049400* (*JAZ9*), and *Solyc03g122190n* (ZIM domain-containing protein, salt responsive protein 1) also belonged to the above group, indicating that JA signalling is involved in regulating high temperature stress tolerance.

2.2.4. GO and KEGG Pathway Enrichment Analysis of Differentially Expressed Genes

Under high temperature stress, a large amount of DEGs involved in different biological processes and functions were observed, including those enriched in cellular process (GO:0009987), metabolic process (GO:0008152), nucleoside binding (GO:0001882), and other terms (Figure 8A). Furthermore, approximately 50 genes involved in organelle organization and biogenesis (GO:0006996), cellular component biogenesis (GO:0044085), and cellular component organization (GO:0016043) were significantly up- or down-regulated after high temperature treatment in samples treated with Spd (Figure 8B). Moreover, under high temperature stress, compared with control, the expression of genes involved in electron transport chain (GO:0022900), NADH dehydrogenase activity (GO:0003954), oxidoreductase activity (GO:0016655), and macromolecular complex assembly (GO:0065003) was dramatically changed (Figure 8C). With the Spd application under normal conditions, genes involved in catalytic activity (GO:0003824) and transferase activity (GO:0016740) were up- or down-regulated (Figure 8D). The GO enrichment analysis suggested that Spd altered the high temperature stress response mode of the plants.

## Hierarchical Clustering of DEGs

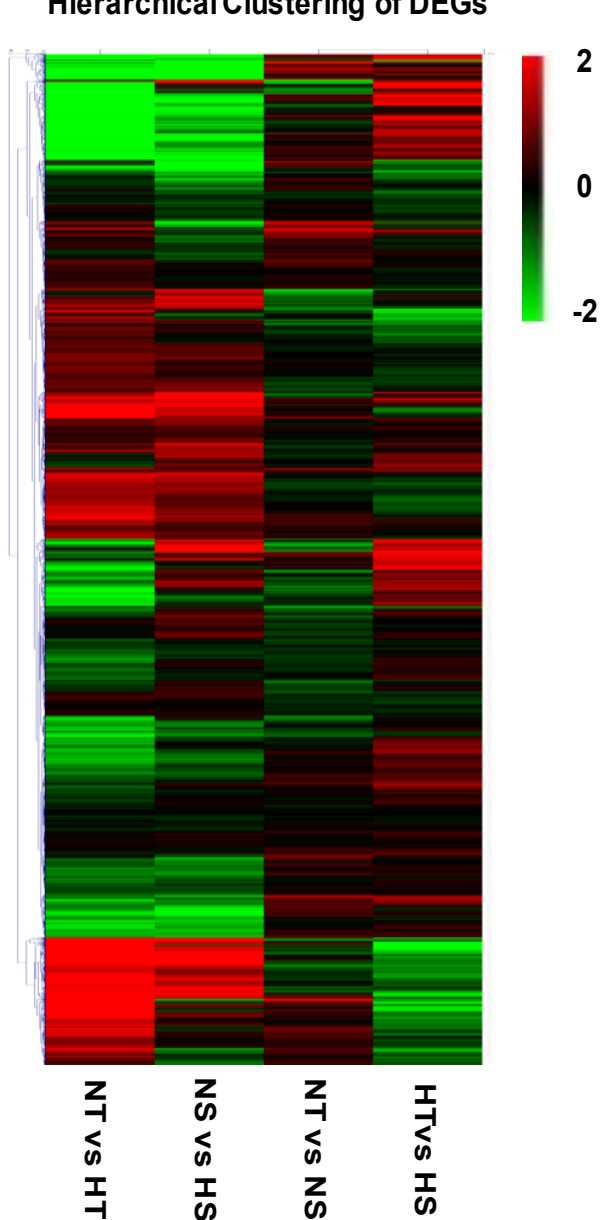

**Figure 8.** Hierarchical clustering of DEGs identified under NT, HT, NS, and HS based on Gene Ontology (GO) terms, showing GO category distribution. NT, normal growth condition under 25/18 °C and water; NS, normal growth condition under 25/18 °C with 1 mM Spd treatment on plants; HT, high temperature growth condition on 38/28 °C and water treatment; HS, high temperature growth condition on 38/28 °C with 1 mM Spd treatment on leaves.

KEGG was used to elucidate the enrichment of genes in NS samples, compared with NT. Five major classes, including cellular processes, environmental information processing, genetic information processing, mentalism, and organismal systems were found to be enriched, divided into 14 pathways (nucleotide metabolism; lipid metabolism; glycan biosynthesis and metabolism; energy metabolism; carbohydrate metabolism; biosynthesis of secondary metabolites; amino acid metabolism; translation; transcription; replication and repair; protein folding, sorting, and degradation; signal transduction; membrane transport; and transport and catabolism); see Figure 9. There were 20 specific biosynthesis and primary/secondary metabolism pathways, as depicted in Figure 9B, including photosynthesis proteins, flavonoid synthesis, amino acids, and betalain biosynthesis.

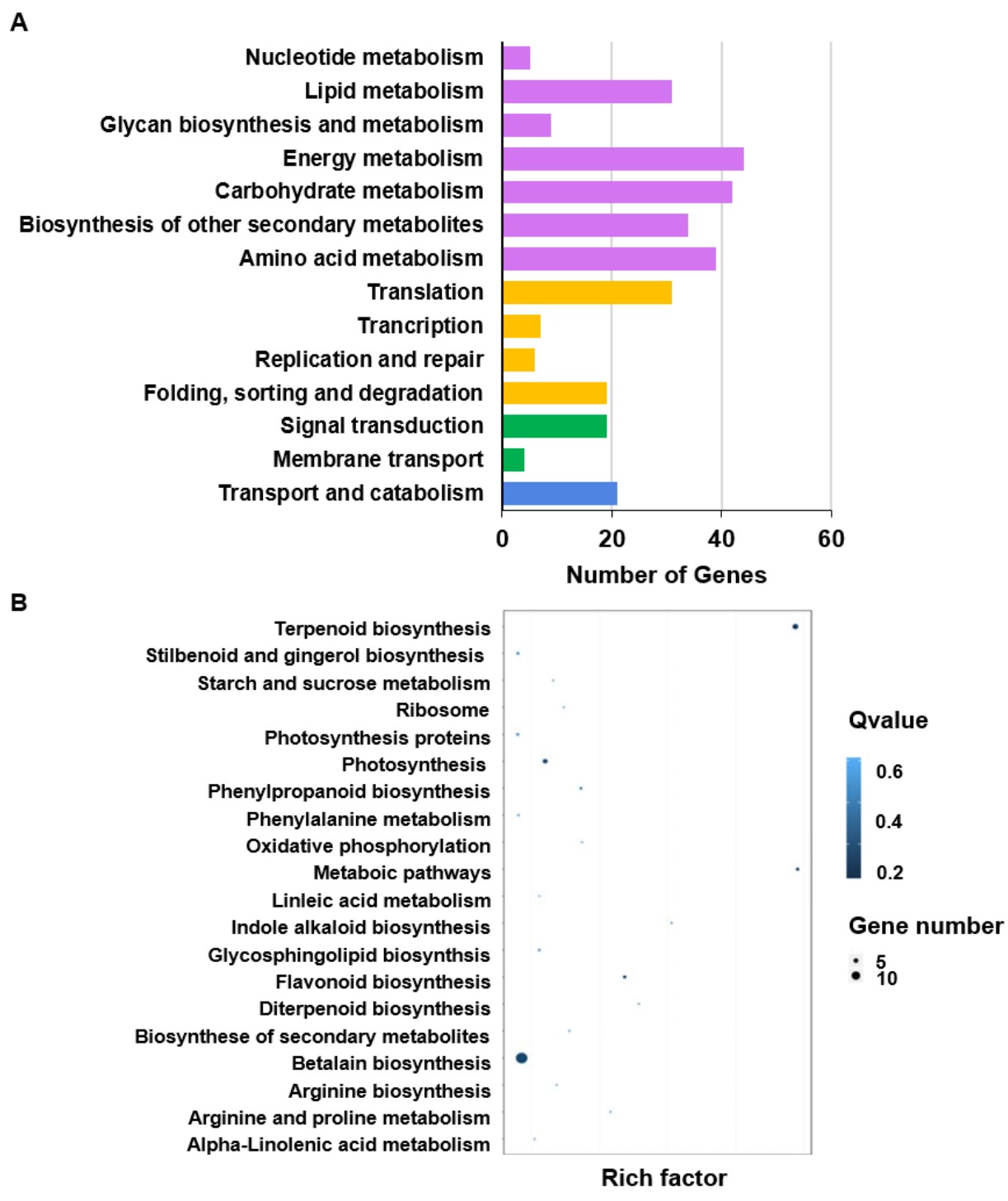

**Figure 9.** KEGG pathway enrichment analysis of DEGs: (**A**) Number of genes; and (**B**) the *y*-axis indicates the enriched pathways, while the *x*-axis indicates the rich factor, which refers to the ratio of the number of DEGs enriched in a certain KEGG pathway to the number of annotated genes. The greater the value, the higher the DEGs enrichment degree. Color indicates q-value and size indicates the DEG count. The chart shows the 20 most-enriched pathways. Note: NT, normal growth condition under 25/18 °C and water; NS, normal growth condition under 25/18 °C with 1 mM Spd treatment on plants; HT, high temperature growth condition on 38/28 °C and water treatment; HS, high temperature growth condition on 38/28 °C with 1 mM Spd treatment on leaves.

### 2.2.5. The Regulation of Energy and Carbohydrate Metabolism Was Closely Related to the Tolerance to High Temperature Stress

The dynamic GO enrichment results revealed that several groups of genes involved in energy metabolism presented obvious expression changes in the HT treatment samples but remained unchanged under the HS treatment. In mitochondria, the energy carrier ATP is generated during electron transfer from NADH to $O_2$ by three large enzyme complexes, composed of NADH:ubiquinone oxidoreductase (NADH dehydrogenase or complex I), ubiquinol:ferricytochorome c oxidorecudtase (cytochrome reductase or complex III) and ferocytochrome c:$O_2$ oxidoreductase (cytochrome oxidase or complex IV) [37]. Among the DEGs, 10 genes encoding NADH dehydrogenase, 3 genes encoding cytochrome oxidases, and 2 genes coding ATP synthase were significantly up-regulated under high temperature stress, suggesting an over-activated energy product for the degradation and biosynthesis of proteins. However, these were all down-regulated in the tomato plants treated with the exogenous Spd under high temperature stress, promoting the stabilization of ATP generation and energy metabolism.

ATP is mainly produced by carbohydrate metabolism, such as glycosis, tricarboxylic acid cycle (TAC), and the pentose phosphate pathway. Thioredoxin activates chloroplastic NADP malate dehydrogenase (NADP-MDH) and fructose-1,6-bisphosphatase in TAC process. In our results, thioredoxin showed different accumulation patterns in response to high temperature; however, the exogenous Spd maintained the thioredoxin expression at a high level. Notably, sucrose synthase (*Solyc07g042520*) and sugar transporter *SWEET1* (*Solyc06g060590*) were down-regulated under high temperature stress, while being restored by the exogenous Spd.

### 2.2.6. Response to High Temperature Related to the Regulation of the Photosynthesis Pathway

Photosynthesis and reproductive development are the physiological stages that are most sensitive to stress [38]. Genes (2 LHCA1C, *LHCA2*, *LHCA3C*, *LHCA4*, *LHCA8*, *LHCA13*, and *LHCA25*, with gene ID *Solyc02g070935*, *Solyc02g071070*, *Solyc03g005780*, *Solyc08g067320*, *Solyc06g069730*, *Solyc12g011280*, *Solyc07g063600*, and *Solyc12g006140*, respectively) encoding sub-units of the light-harvesting chlorophyll–protein (LHC) complex, which absorbs light and passes it to the light reaction center of the corresponding photosystem, showed decreased transcript abundance under high temperature stress. Among these, the exogenous Spd increased the expression of *LHCA2, LHCA4,* and *LHCA25*.

### 2.2.7. Epigenetic Genes Display Distinct Expression Profiles in the Heat-Treated and Heat/Spd-Treated Tomato Plants

Epigenetic factors, such as DNA methylation and histone modifications, are significant in gene expression regulation when plants encounter disadvantageous environmental conditions. Heat stress leads to the transient activation of repetitive elements or silenced gene clusters close to the centromeric regions, as well as the transient loss of epigenetic gene silencing [39]. Some genes associated with epigenetic regulation were up-regulated in response to high temperature, including histone demethylase (*Solyc03g121930*), histone deacetyltransferase (*HDAC*, *Solyc02g089790*), and RNA-dependent RNA polymerase 6 (*RDR6*, *Solyc04g014875*). The increased accumulation of histone demethylase, *HDAC*, or *RDR6* might cause transcriptional or post-transcriptional repression of heat-responsive genes. Notably, histone–arginine methyltransferase (*Solyc05g054240*), a transcriptional activation factor—the expression of which may be repressed by high temperature—was induced by Spd under heat stress. This gene may be involved in the transcriptional activation of heat-responsive genes.

### 2.2.8. The Response of Detoxification Signaling to High Temperature Stress Was Different in the Control and Exogenous Spd Treatment

Most of the damage induced by heat stress has been identified as part of detoxification signaling caused by stress injury, protein denaturation, and excessive accumulation of reactive oxygen species (ROS) or nitric oxide (NO), initiating corresponding events such as the expression of heat response genes, including (but not limited to) molecular chaperones helping protein folding and maturation, proteinases that remove denatured proteins, and the activation of genes encoding enzymes involved in the generation and removal of ROS and other detoxification proteins.

ROS and NO exert multiple modulating effects on abiotic stress responses. Low concentrations of ROS and NO act as small signaling molecules for inducing tolerance to stress; however, large amounts of ROS and NO can be toxic and repress the growth of plants. Excessive doses of ROS and NO cause cellular damage, as membrane lipid and pigment peroxidation compromise membrane permeability and function [1]. In our data, nitric oxide synthase (*Solyc01g006180*) was significantly increased in expression under high temperature for NO production, but showed a 50% reduced increase in Spd-treated samples, in comparison. Phosphomethylethanolamine N-methyltransferase (PEAMT), a rate-limited enzyme involved in betaine and phosphatidic acid synthesis, is also involved in the regulation of osmotic balance and oxidative damage caused by abiotic stress. The PEAMP expression level was tremendously down-regulated in response to heat stress, but was restored by Spd. Glutathione S-transferase/peroxidase (GST) plays a role in the detoxification of steroid metabolism and modulating oxidative stress by eliminating 4-hydroxynonenal. A gene coding GST (*Solyc07g056480*) was repressed by high temperature, but inversely was induced by the exogenous Spd.

Protein folding is critical for their maturation and stabilization. Heat stress hampers protein translation, resulting in imperfect proteins that will be led to degradation. Heat shock proteins (HSPs) are typically induced when cells are exposed to high temperature stress, and are closely related to protein folding, maturation, and stabilization. HSPs operate as a complex with other components, called co-chaperones. Three genes annotated as BAG family molecular chaperones—which are co-chaperones for the HSP70 and HSC70 chaperone proteins—were down-regulated in HT samples, but normally expressed in HS samples.

### 2.2.9. The Regulation of Flavonoid Synthesis and Cell Wall Related Genes Was Associated with Spd Mediated Tolerance to Heat Stress

The KEGG analysis results revealed several groups of genes involved in flavonoid synthesis and cell wall biogenesis that presented opposite expression patterns in the HT and HS samples. As important secondary metabolites, flavonoids widely exist in plant leaves, flowers, fruits, and other tissues. The KEGG pathway analysis showed that the DEGs sensitive to high temperature were significantly enriched in the flavonoid biosynthesis pathway, including one gene annotated as flavonol synthase (*Solyc03g096050*), four genes encoding 4-coumarate–CoA ligase (*Solyc06g068650*, *Solyc03g111170*, *Solyc03g117870*, and *Solyc12g094520*) for the catalysis of phenylpropanoid-derived compounds, including anthocyanins, flavonoids, isoflavonoids, coumarins, lignin, suberin, and wall-bound phenolics, which were induced by Spd under heat stress, but suppressed by high temperature.

Similarly, among the DEGs in HT and HS treatments, one gene encoding cinnamate 4-monooxygenase, which controls carbon flux to pigments essential for protection against numerous phytoalexins synthesized by plants when challenged by stress, as well as lignins, was also repressed by HT, but normally expressed in HS samples. These genes participate in lignin biosynthesis and promote the cell wall flexibility under high temperature. Two Xyloglucan endotransglucosylase/hydrolase (XTH) genes, which cut and re-join hemicellulose chains in the plant cell wall, thus contributing to wall assembly and growth regulation, were specifically induced by Spd under high temperature. One bifunctional wax ester synthase/diacylglycerol acyltransferase (*WSD1*, *Solyc01g095930*), which is in-

volved in cuticular wax biosynthesis, cellulose synthase (*Solyc12g014430*), was specifically induced by Spd under high temperature, indicating the function of exogenous Spd in inner and cuticular cell wall reinforcement.

### 2.3. Role of Transcription Factor MYC2 in Different Heat Tolerance Varieties

In order to verify the difference between heat-resistant and non-heat-resistant varieties at the molecular level, MYELOCYTOMATOSIS ONCOGENE HOMOLOG 2 (MYC2) and JASMONATE-ZIM-PROTEIN (JAZ1) were selected, based on the RNA-seq results, and subjected to quantitative reverse transcription PCR (RT-qPCR) analysis in tomato leaves treated with Spd under normal and high temperature conditions. From the results, the activation and expression of MYC2 in five varieties after high temperature treatment were obtained (Figure 10A). The heat-resistant cultivars (Shenfen 16 and Puhong 968) were all activated and expressed, but there was no change in the expression level of non-heat-resistant cultivars (sufen-9 and shenfen-19). As MYC2 is known to be a master regulator of JA signaling and plays a significant role in plant defense against heat stress, the expression result suggested that the heat-resistant tomato cultivars may rely on MYC2 to transmit heat conduction stress signals. As for JAZ1, the negative regulatory factor of MYC2, its expression level was also increased in both of the heat-resistant tomato varieties Shenfen 16 and Puhong 968 (Figure 10B), from which we can assume that the increased expression of JAZ1 is used to balance the JA heat resistance signal mediated by MYC2, thus inhibiting the transcriptional activation activity of MYC2 transcription factor at the protein level. In contrast, JA activation was not obvious in the non-heat-resistant tomato varieties (Shenfen 9 and Shenfen 19), and the expression of MYC2 and JAZ1 was also unchanged.

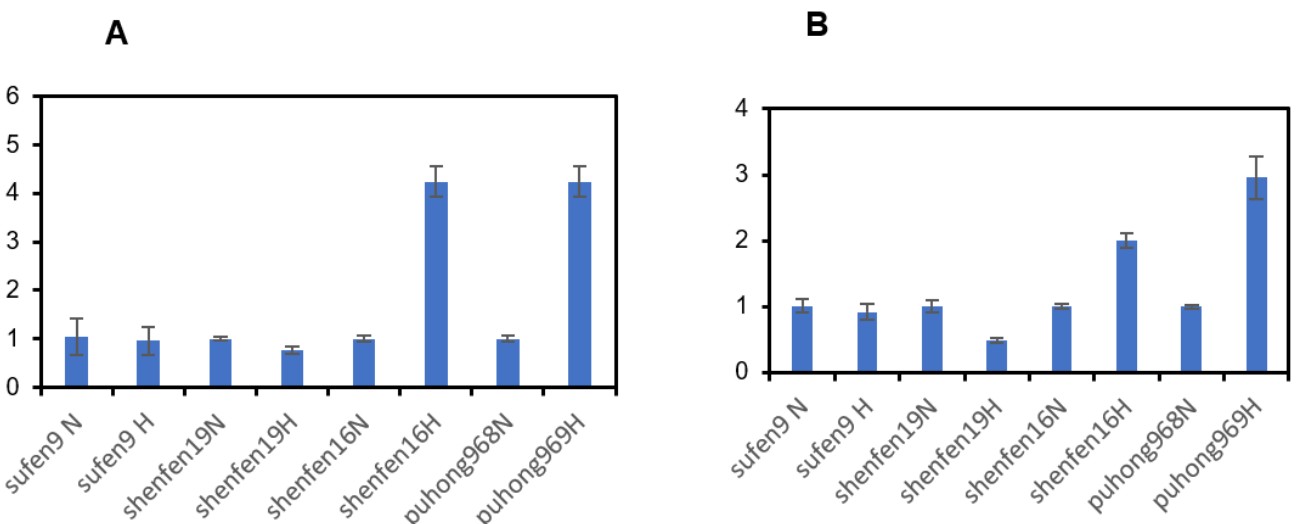

**Figure 10.** MYC (**A**) and JA (**B**) activation in different types of samples. Note: NT, normal growth condition under 25/18 °C and water; NS, normal growth condition under 25/18 °C with 1 mM Spd treatment on plants; HT, high temperature growth condition on 38/28 °C and water treatment; HS, high temperature growth condition on 38/28 °C with 1 mM Spd treatment on leaves.

### 3. Discussion

#### 3.1. The Heat Stress Response Mechanisms under Exogenous Spd

Tomato is relatively sensitive to high temperature, and its growth is constrained under heat stress, especially in greenhouse cultivation in summer. In this study, the Spd solution- and water-treated tomato plants presented distinct patterns of heat tolerance under heat stress. The growth rate, dry weight, and fruiting rate of plants treated with exogenous Spd were all higher than controls under high temperature, possibly due to photosynthesis, protein and hormone metabolisms, and signaling pathways being affected by exogenous Spd. The Spd-treated plants presented more robust transpiration and photosynthetic capabilities than controls under high temperature, and showed a normal intercellular $CO_2$ concentra-

tion, suggesting an undamaged photosynthetic system in HS samples. Congruously, in HS samples, chlorophyll and carotenoid contents were not decreased, compared with NT. Zhang et al. [40] reported that exogenous Spd had a positive effect on the chlorophyll $\alpha$ fluorescence transients, and enhanced the heat tolerance of tall fescue by maintaining cell membrane stability, increasing antioxidant enzyme activities, improving photosystem II (PSII) and relevant gene expression. Similar results have also been reported in other plants, such as cucumber [41], rice [42], white clover [43], wheat [44], creeping bentgrass [45], cauliflower [46], and lettuce [47].

The soluble protein contents were significantly increased in HT samples, showing a typical cell response to stress; however, this parameter was not changed in Spd-treated tomato plants, indicating that Spd signaling might inhibit heat stress-triggered soluble protein accumulation. Zhang et al. [48] also reported that exogenous Spd increased soluble protein contents in tomato leaves under drought stress conditions.

In this study, GO analysis of the obtained transcriptome data showed that DEGs associated with the cellular process, metabolic process, electron transport chain, and oxidoreductase activity were mostly changed by Spd under heat stress. Based on the transcriptome, phenotypic, and physiological characteristics of the control and Spd-treated tomato plants, we speculated that the enhanced drought tolerance meditated by Spd is possibly due to a robust photosynthetic capability and optimized metabolism system under high temperature. The differentially expressed genes identified in other transcriptome analyses related to high temperature stress after Spd application have also indicated that Spd can enhance heat tolerance by promoting photosynthetic capacity. For example, of 6697 differentially expressed genes, highly enriched terms included an oxidation–reduction process for biological process (BP), the photosystem II oxygen evolving and thylakoid membrane for cellular component (CC), and catalytic activity, oxidoreductase activity, and hydrolase activity for molecular function (MF) [49].

### 3.2. Energy, Carbohydrate Metabolism, and Photosynthesis Process Presented Diverse Responses in HT and HS Samples

Heat stress adversely impacts many aspects of the physiology of plants, especially photosynthetic capacity and respiration [1,33,35]. The early effects of heat stress involve structural alterations in chloroplast protein complexes and reduced activity of enzymes. Some genes in the photosynthesis pathway showed decreased transcription abundance in HT samples while, as expected, the opposite response was observed in HS samples. DEGs encoding LCHA showed decreased expression in HT, which indicated that the absorption of light energy and the resulting photosynthetic electron transport process might have been limited under heat stress. In a previous study, the contents of PSI component ferredoxin-NADP reductase, PSII component oxygen-evolving enhancer proteins, and chlorophyll biosynthesis proteins were all tremendously induced by Spd under heat stress [32], while not being changed in the transcriptome data. This hints that Spd might stabilize and promote protein expression at a post-transcriptional level, potentially explaining why the chlorophyll accumulation level increased, even though synthesis-associated genes were down-regulated in HS samples (Figures 3 and 4).

Photochemical modifications in the carbon flux of the chloroplast stroma and those of the thylakoid membrane system are considered the primary sites of heat injury, as photosynthesis and enzymes of the Calvin–Benson cycle, including ribulose 1,5-bisphosphate carboxylase (Rubisco) and Rubisco activase, are very sensitive to increased temperature and are severely inhibited even at low levels of heat stress [5]. In a previous study, exogenous Spd had positive effects on Rubisco and RCA in tomato leaves, helping to maintain the Calvin cycle and photosynthetic carbon assimilation at high levels [32]. In this study, Calvin cycle genes were not differentially regulated by HT and HS treatment at the transcriptional level, suggesting that Spd promotes photosynthetic carbon assimilation through direct regulation at the post-transcriptional level. On the other hand, Spd maintained the ATP

synthesis balance by affecting ATP biosynthesis gene expression, including TAC and ATP synthase genes.

Sang et al. (2017) [32] evaluated the functions of exogenous Spd in high temperature stress responses in tomato leaves through proteomic analysis and identified a total of 67 differentially expressed proteins, where the four largest categories included proteins involved in photosynthesis (27%); cell rescue and defense (24%); protein synthesis, folding, and degradation (22%); and energy and metabolism (13%).

Taken together, these findings provide a better understanding of the Spd-induced high temperature resistance through proteomic approaches, providing valuable insight into improving high temperature stress tolerance, which is expected to become increasingly important throughout the global warming epoch.

### 3.3. Transporter Genes Are Less Affected by Heat Stress upon Spd

Biomembrane transporters are critical for plant development and stress adaption. In heat-treated tomato plants, the phosphate transporter gene *PHO1*, two amino acid transporter genes, an ammonium transporter gene, a sulfate transporter gene, and the exocyst complex component *EXO70A* were all down-regulated, resulting in blocked transport of phosphorus, amino acid, ammonium, sulfate, and some secretory proteins. These small molecular materials are essential for plant growth or fruit ripening, where deficiency of phosphorus results in dwarf plants, delayed flowering, and wizened seeds, while insufficient nitrogen causes failed fruit expansion. The expression of the above genes was not as affected in HS samples, compared with HT samples.

Aquaporins are membrane channels that facilitate the transport of water across the biological membranes of most living organisms. In plants, aquaporins promote water transport into the cellular lumen and play an important role in water homeostasis through turgor regulation. Previous studies have revealed that aquaporin-like proteins—also called major intrinsic proteins—are divided into the PIP, TIP, and NIP subfamilies. Three TIP and PIP genes showed significantly decreased transcript abundance in HT samples; however, these were unchanged or only mildly decreased in HS samples. This result suggests that a general down-regulation of AQPs might minimize water flow through cell membranes and uphold leaf turgor in heat-tolerant samples, while an excessive decrease in AQPs could cause too high leaf turgor, thus inhibiting the normal plant cell physiological processes.

### 3.4. Differentially Expressed Genes Implicated in ABA and JA Response

We considered hormone signaling pathways involved in the plant tolerance to abiotic stress. ABA, salicylic acid (SA), and ethylene (ET) present increased levels under heat stress, while others decrease, such as cytokinin (CK), auxin (AUX), and gibberellic acid (GA) [1]. Late embryogenesis abundant (LEA) proteins have been found to play important roles in protection against heat and drought stress, which are also induced by ABA [1]. In the classification of ABA, the ABA-responsive element binding factor (AREB-like protein) and LEA genes are induced by high temperature. Here, *LEA* genes were significantly induced by heat stress. However, among the DEGs, ABA 8′-hydroxylase, known for catalyzing the key step of ABA catabolism of hydroxylation at the 8′-position of ABA [50], was down-regulated in the HS sample, compared with HT. As is well-known, over-activated ABA signaling can seriously inhibit plant growth. Therefore, Spd might restore plant growth by ameliorating the increase in ABA content. All of the evidence here suggests that Spd-triggered heat stress tolerance might not be dependent on the ABA signaling pathway. As for the JA signaling pathway, allene oxide cyclase (AOC)—a precursor of JA—and MYC2—a positive regulator of JA—were down-regulated by heat stress but normally expressed under Spd treatment. Meanwhile, JAZ—a suppressor of MYC2—was not changed under high temperature, but was induced by Spd. These results suggest that JA signaling might be involved in the heat stress responses specially regulated by Spd.

## 4. Materials and Methods

### 4.1. Plant Materials and Treatment

Tomatoes (*Lycopersicon esculentum* Mill. cv. Puhong 968) were germinated and grown in sterile pots with growth substrate. This paper continues the previous experiments of the author's laboratory (Key Laboratory of Southern Vegetable Crop Genetic Improvement, Ministry of Agriculture), and we continued to use the tomato varieties previously screened out by Su [51]. Similarly, the temperature environment regulation also followed the previous experiment. The seedlings were grown under 14 h light and 10 h dark at 25/18 °C, relative humidity of 55–65%, and light intensity of 400–800 μmol m$^{-2}$·s$^{-1}$ in the growth chambers (Ningbo Jiangnan Instrument Factory, Ningbo, China).

Tomato seeds were seeded in pots for seedling cultivation. When the plant had two leaves and one shoot, those with the same growth trend were selected and transplanted to a small pot and placed in a light incubator for cultivation. When the fourth true leaf had developed and fully expanded, plants of each variety were taken and placed in the light incubator. The seedlings were separated into two groups, which were, respectively, treated with distilled H$_2$O or 1.0 mmol·L$^{-1}$ Spd sprayed on leaves. After 2 days, the seedlings were subjected to high temperature (38/28 °C, day/night; light/dark photoperiod, 14/10 h; relative humidity, 55–65%) for 7 days. There were four different treatments, including NT (normal treatment, 25/18 °C, day/night; light/dark photoperiod, 14/10 h; relative humidity, 55–65% + water), NS (normal condition + Spd), HT (high temperature 38/28 °C, day/night + water), and HS (high temperature + Spd). The application of exogenous Spd or water was carried out at 17:00 every day. The fourth leaves were used for further determination of indices and transcriptome sequencing.

In the test of role of the transcription factor *MYC2* in different heat tolerance varieties, further tomato varieties were selected, including Shenfen 16, Puhong 968, Sufen 9, and Shenfen 19. These four tomato varieties were collected from Shanghai Academy of Agricultural Sciences. Their treatment method is detailed in Section 4.9 below.

### 4.2. Determination of Plant Growth

Tomato height was measured with a ruler, and dry weight was assessed by weighing whole plants or aerial parts after drying at 75 °C in an oven for 72 h. For tomato production assay, 320 plants were divided into four groups, and every group was performed with four different treatments (as a normal condition and high temperature condition, each with or without exogenous Spd). Then, the treated plants were transferred to the greenhouse for fruit set testing. The fruit set rate was calculated by dividing the fruiting number by the flowering number.

### 4.3. Measurement of Chlorophyll Content and Photosynthetic Parameters

Chlorophyll and carotenoids were extracted with organic solvent (acetone, ethanol, and water, 4.5:4.5:1 by volume), and the content was measured according to the method of [52].

The net photosynthetic rate (PN), stomatal conductance (Gs), transpiration rate (Tr), and intercellular CO$_2$ concentration (Ci) of the fully expanded leaves (i.e., the third leaves from the base of the stem) were measured using a portable photosynthesis system (LI-6400, LI-COR Inc., Lincoln, NE, USA), with leaf temperature, RH in the assimilation chamber, external CO$_2$ concentration, and light intensity maintained at 25 °C, 70%, 380 $\pm$ 10 μmol mol$^{-1}$, and 1000 μmol photons m$^{-2}$s$^{-1}$, respectively.

### 4.4. Assay for Soluble Sugar and Soluble Protein Contents

For the soluble sugar contents assay, fresh samples were ground in 95% (*v/v*) ice-cold methanol. The materials were analyzed with HPLC, using the method described by [53]. Soluble protein was extracted with buffer solution (30 mmol·L$^{-1}$ Tris-HCl, pH 8.7; 0.7 mol·L$^{-1}$ sugar; 1 mmol·L$^{-1}$ DTT; 1 mmol·L$^{-1}$ AsA; 1 nmol·L$^{-1}$ EDTA; 1 mmol·L$^{-1}$ MgCl$_2$; and 1 mmol·L$^{-1}$ PMSF), and the contents were measured using the G250 method.

### 4.5. RNA Extraction and Sequencing

RNA extraction was conducted using TRIzol reagent (Invitrogen, Carlsbad, CA, USA). The quality of the RNA was confirmed by agarose gel electrophoresis and quantified with a NanoDrop 2000 spectrophotometer (Thermo Scientific, Vilnius, Lithuania). After treatment with DNase I, the mRNA was isolated using a Dynabeads mRNA Purification Kit library was constructed following the manufacturer's instructions of the Truseq RNA Sample Prep Kit (Illumina, Woodlands, Singapore). Briefly, the enriched mRNA was first fragmented with $MgCl_2$ solution, following which the first- and second-strand cDNA were synthesized using a random primer. In the next step, 3′-ends of cDNA were adenylated and 5′-ends were repaired. Then, the cDNA fragments were ligated with DNA sequencing adapters for PCR amplification. Finally, the constructed cDNA libraries were sequenced on a flow cell using an Illumina Hiseq$^{TM}$ 4000 platform (Illumina, San Diego, CA, USA). During QC, an Agilent 2100 Bioanalyzer and an ABI StepOnePlus Real-Time PCR system were used for the quantification and qualification of the libraries. A total of four libraries were constructed, where each library was represented by three repeats.

### 4.6. Reads Mapping

Low-quality reads, as well as those containing adapters and high content of unknown bases were removed before downstream analyses. After filtering, the raw reads were mapped to the genome version SL2.50 downloaded from the NCBI (https://www.ncbi.nlm.nih.gov/assembly/GCF_000188115.3, accessed on 10 August 2022) using HISAT [54].

### 4.7. Novel Transcripts, SNP/INDEL, and Differentially Splicing Genes Detection

After genome mapping, we used StringTie [55] to reconstruct transcripts and, with genome annotation information, we identified novel transcripts existing in all libraries using cuffcompare, a tool in cufflinks [56]. The SNP and INDEL variants for each library were called with GATK [57]. rMATS was used to detect differentially splicing genes (DSG) between samples. The statistical model of MATS calculates the *p*-value and false discovery rate (FDR). A difference in the isoform ratio of a gene between two conditions (in our project, genes FDR $\leq$ 0.05) defined a differentially splicing gene (DSG).

### 4.8. Differentially Expressed Genes Analysis and Gene Function Annotation

For the four different treatment samples, DEGs were analyzed using the DESeq 2 [58] R package, with three replicates per sample. A false discovery rate (FDR) *p*-value less than 0.05 and a threshold fold change ($\log_2$) $\geq$ 2 were used to identify differentially expressed genes between two samples. The identification of unique or overlapping genes within the different expression data of the samples and the generation of Venn diagrams were carried out using Draw Venn Diagram (Genomics). Heatmaps and hierarchical clustering analyses were produced in Multiple Experiment Viewer (MeV) version 4.9.0 [59]. Gene Ontology (GO) term annotation and GO enrichment for differentially expressed genes were obtained by online analysis using agriGO version 2.0 [60]. The GO enrichment was completed under a significance threshold of 0.05 following the Hochberg FDR correction.

### 4.9. Role of Transcription Factor MYC2 in Different Heat Tolerance Varieties

To understand the role of the transcription factor *MYC2* in different heat tolerance varieties, an experiment using different varieties was carried out in this study. Two heat-resistant varieties (Shenfen 16 and Puhong 968) and two non-heat-resistant varieties (Sufen 9 and Shenfen 19) were used for this experiment. The above four tomato varieties were seeded in the plug, and each variety was planted with 30 plants. When the plants grew to two leaves and one bud, 20 tomato seedlings with the same growth vigor were selected from each variety, transplanted into a small pot, and cultivated in a light incubator. When the leaves grew to three leaves and one bud, ten plants were taken from each variety and placed in the light incubator. The culture conditions were as follows: light intensity 4000 LX, light time 12 h/d. It was heated to 40 °C at 10:00 a.m. every day for four hours, then cooled to

28 °C and held to 10:00 the next day. The treatment lasted for 3 days. The other ten plants of each variety were treated with a temperature of 28 °C all day and night as control. The leaves of each variety treated with normal temperature and high temperature were evenly sampled for further analyses.

First, total RNA was extracted from plants, as above. Then, reverse transcription of mRNA was determined. The following reaction solutions were prepared in a 200 μL centrifuge tube: Total RNA 2 μg, 0.5 M oligo (DT) 18 1 μL, and DEPC $H_2O$ to 12 μL. Then, denaturation was carried out at 75 °C for 10 min, followed by cooling on ice for 2 min. The following reagents were added: 10ded:10ed in 20μL, dNTP mixture (2 mm each) 1μL, RNase inhibitor (0.25 U /μL) 0.5 μL, m-mlv 1μL, and DEPC $H_2O$ to 20μL. The reverse-transcribed cDNA was obtained at 42 °C for 1 h and 75 °C for 15 min. Finally, qRT-PCR was performed for each treatment. Primer design for RT-PCR was as follows: TGGTCGGAATGGGACAGAAG(slactin-F), CTCAGTCAGGAGAACAGGGT(slactin-R). Next, we prepared the fluorescent quantitative PCR reaction solution. We added 2 TG-GTmplate cDNA for each tube, and centrifuged to force the reaction solution to gather at the bottom of the tube and cover the membrane. The reaction was performed on ABI fluorescence quantitative PCR, according to the following procedure: 95 °C for 10 min, 95 °C for 15 s, 60 °C for 1 min, read the fluorescence value, go to step 2, repeat for 40 cycles. The dissolution curve was carried out as follows: 95 °C for 10 s, 65 °C for 1 min, 0.2 °C step temperature increase, read fluorescence value, wait until the temperature rises to 95 °C.

## 5. Conclusions

The results obtained in this study showed that exogenous Spd can promote the growth of tomato plants, as well as their tolerance to heat stress. This might involve Spd regulating the following processes: (1) photosynthetic capacity, energy, and carbon metabolism; (2) cell wall remodeling; (3) transport of nutrient molecules, secretory proteins, cations, and water; (4) cell rescue and detoxification; (5) transcriptional activation and de-activation controlled by epigenetic components; and (6) hormone signaling homeostasis.

**Supplementary Materials:** The following supporting information can be downloaded at: https://www.mdpi.com/article/10.3390/agronomy13020285/s1, Figure S1: Summary of gene expression of each sample.

**Author Contributions:** Conceptualization, C.P. and S.G.; validation, C.P.; writing—original draft preparation, C.P.; writing—review and editing, S.S., Y.W., L.S. and Jahan, M.S.J.; investigation, C.P. and J.D. All authors have read and agreed to the published version of the manuscript.

**Funding:** This research was funded by National Natural Science Foundation of China, grant number 32272793 and China Agriculture Research System, grant number CARS-23.

**Data Availability Statement:** The data supporting the findings of this study are available from the author Chen Peng and corresponding author Prof. Shirong Guo, upon request.

**Acknowledgments:** We would like to acknowledge Sun Jin for his technical guidance, as well as the doctoral and master students from Key Laboratory of Southern Vegetable Crop Genetic Improvement who offered technical assistance, such as Su Xiaoqiong, Shanxi, Sang Qinqin, Wei Ying, Xu Qing, Yue Dong, Zhu Heyuan, Zhou Xinpeng, Zhuang Yanrong, Liu Bo, Wang Siyi, and Cao Xiaomeng, among others.

**Conflicts of Interest:** The authors declare no conflict of interest. The funders had no role in the design of the study; in the collection, analyses, or interpretation of data; in the writing of the manuscript; or in the decision to publish the results.

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
