# Peer review of "Transcriptome Analysis of the Regulatory Mechanism of Exogenous Spermidine in High Temperature Stress Resistance of Tomato Seedlings"

_agronomy, doi:10.3390/agronomy13020285_

Round 1

Reviewer 1 Report

Although the experiment involved a huge work to obtain updates on the heat stress related mechanism in tomato, the manuscript has many flaws and should be improved. In addition, the English language, that made the overall reading difficult, should be revised.

Abstract: Many sentences are not clear, both in the grammar form or in the scientific meaning.

Introduction:

Lane 44: the sentence is not clearly written since it seems that the QTL analysis at genomic level is a tool for increasing plant tolerance to heat stress that sounds not correct.

Lane 55 and lane 58: the sentences are not clear

Results

Lane 67: is not clear…something is missed

Lane 69: The authors indicate that Sp treatment has a positive effect on fruit setting under HT tolerance, but Sp increase this parameter also in normal condition and at higher rate. The authors should check this sentence based on the Figure 1D

Lane 85: The sentence is not clear, the values reported regard the NT vs HT condition. What I understand is that the Sp acts in the same way in normal and high temperature condition.

Lane 139: From the text I understand that the 35833 novel transcripts include the others, but if you sum all the values reported, in the text and in the table 3, (from novel coding transcript to novel gen), the total amount is different.

Lane 184: I guess that the supplementary file is 1B not 2B

Lane 210: It is difficult understand the number of upregulates and downregulates genes considering the text and the Figure 7A

Lane 215 – 229: The explanation of the genes involved in the tolerance and the Sp treatment are not clear

Lane 297: …” decrease expression of…. could be elevate” ….is difficult to understand properly

Lane 362 – 385: All the paragraph should be re written since the results of MYC2 gene expression and the JA involvement are not well explained

Discussion: The discussion is not focused on the explanation of the results, but it seems a copy of the results chapter. The authors should improve the discussion by trying to explain the results obtained also reporting other works from literature.

Material and methods

All the experimental design is not well explained. It seems that different experiments were made. In addition is not clear if the plants form which the RNA was collected for the RNAseq were used also for some parameters detection (the no destructive ones). In same case the authors divided the plants in two groups and in others in four groups.

Lane 488: the HT conditions are missed

Author Response

Please see the attachment, thank you!

Reviewer 2 Report

The paper "Transcriptome Analysis on the Regulatory Mechanism of Exogenous Spermidine in Tomato Seedlings resistance to High Temperature Stress" describes an interestin application ff spermidine facing the climate changes. I have some doubt, regarding the experimental design and the quality of presentation.

General considerations:

- The numbers from one to nine must be written with letters

- Be careful in the use of the font, the text alignment, character dimension and generally in the final formatted text. There is a lot of errors.

Abstract. I think that the last sentence must be changed and/or modified with a stronger conclusion.

Introduction is well described and fits adequately with the focus of the paper. In lane 39: please define the plasma membrane before, and the acronym in the brackets.

The phenotypic results are well described and are robust and clearly evidences the role of Spd in the heat stress tolerance over different biochemical aspects. Regarding the part of the RNA sequencing, I don't understand the reason of the SNPs and indels analysis. Have you used different varieties for the experiments? I understand that you used heat-tolerant and non-tolerant varieties, but it is not clarified in the material and methods. Then, some part of the paragraph "differentially splicing gene detection" (that lacks of the paragraph number), could be splitted in material and methods (lane 174-177) and discussion (lane 171-174). The description of genes, GO terms and so on is good.

Paragraph 2.2.9 Can you clarify better the role of Spd under heat stress? Is the activity of the Spd suppressed from high temperatures in the flavonoid synthesis?

paragraph 2.3 How have you defined the varieties as heat-not-resistant and heat-resistant?

Discussion. In some part seems a summary of the results without an appropriate bibliography that could help to evidences of your work. In the first paragraph ("The Heat stress [...] Exogeonous Spd"), could be useful compare your work with the available state-of-art . Analogously for the paragraph "Transporter Genes are Less Affected by Heat Stress upon Spd"

Material and Methods. Clarify better the used varieties. 

Lane 481 I don't understand what is the "heart" in plant. Please let me now.

The last thing. Why have you used the SL2.50 version for the alignment of the read? This is not the last one version.

Author Response

Please see the attachment, thank you!

Round 2

Reviewer 2 Report

The coverletter and the manuscript clarified and satisfy my doubts, so I accept the paper in the present form.